# OpenReview forum: "Uncovering and Quantifying Social Biases in Code Generation"
_NeurIPS.cc/2023/Conference — NeurIPS 2023 poster_

### Official Review · Reviewer_y9do · 2023-06-23

**Soundness:** 3 good
**Presentation:** 3 good
**Contribution:** 3 good
**Rating:** 6
**Confidence:** 3

**Summary:**

This paper investigates the problem of social biases in the code generation procedure. This is the first paper to study the social bias issue in the code-generating model and demonstrate the existence of social biases. The author also introduces some evaluation metrics to quantify the bias in code as well as a classifier for scoring. They also conduct many experiments and ablation studies to provide more insight into this direction. In addition, they also validate their result and method by human annotation.

**Strengths:**

This paper is the first paper to study social bias in code generation model.

The result are also interesting that since the training datasets are often human-irrelevant but it still shows that the model will contains biases.  It opens up new avenues for research and discussion about the nature of bias and how it can be subtly embedded in models.

Also the human labeled prompt and dataset will be a great value for future research

It conduct pretty solid experiments on various models and also conduct human annotation to further evaluate the results.

**Weaknesses:**

The authors of this paper have taken a novel approach to uncovering biases in code generation models by constructing biased prompts. While this methodology is sound, the results are somewhat predictable. Given a biased prompt, it is likely that a code generation model will produce biased output. However, this is more a reflection of the input rather than an inherent bias within the model. For instance, a model generating 'find_good_people == 'Hispanic'' is logically equivalent to generating 'find_delicious_apple = red', but we wouldn't label the latter as biased. The model is not able to realize this unless it is trained to recognize certain attributes or words as sensitive.

The study primarily focuses on 'direct bias', where the model is intentionally prompted to generate biased code. In real-world applications, it's unlikely that developers would intentionally craft such biased prompts unless they specifically aim to generate biased code. My main concern lies with 'unintentional bias', where biases are generated even when the prompt has no apparent bias. This is a more pressing issue in real-world scenarios.

Furthermore, it's worth considering that code may appear unbiased at the point of generation, but could introduce bias when applied in certain contexts. This is a complex issue that warrants further investigation.

Another weakness of the paper lies in its oversimplification of certain aspects in its analysis and evaluation. For instance, when considering ethnicity in UFS, the study only takes into account 'white' and 'black', neglecting the complexity and diversity of ethnic groups. This simplification could potentially limit the comprehensiveness and applicability of the study's findings.

I do appreciate that this study represents an initial step towards addressing the question of bias in code generation models. However, future research should aim to delve deeper into the nuances of unintentional and context-dependent biases to provide a more comprehensive understanding of this issue.

**Questions:**

1. In line 135, you mentioned you have 5 types of modifiers and 8 types of demographic dimensions, how did you come up with the prompt dataset with 392 samples?

2. In Table 3, why you don't include the statistics of Codex?

 3. It seems like in line UFS can only applied to binary demographics, is it possible to expand this so that it can be used in ethnicity where there are more than two category?





**Limitations:**

In the related work part, might need to include more papers that study social bias in NLP and the evaluation methods and benchmarks.

---

> ### Author Rebuttal · Authors · 2023-08-09
>
> We would like to first thank the reviewer for the valuable and insightful comments!
>
> **Q1: The authors of this paper have taken a novel approach to uncovering biases in code generation models by constructing biased prompts. While this methodology is sound, the results are somewhat predictable. Given a biased prompt, ...... The model is not able to realize this unless it is trained to recognize certain attributes or words as sensitive.**
>
> **A1**: We agree with you that "the model is not able to realize this unless it is trained to recognize certain attributes or words as sensitive". This provides the inspiration for developing a debiasing method. However, we realize that there may be some misunderstandings here. (1) We would like to attract attention to the research premise of our work, which is written in Line 74-77. As is written, the topic of social bias is only meaningful under human-relevant scenarios. Therefore, we don't view "find_delicious_apple=red" as biased because it is human-irrelevant. (2) We also want to clarify that our prompt is not biased. As shown in Figure 2, the first two functions in our prompt are human-irrelevant, and the third function name only contains a modifier and a human-relevant attribute. The prompt does not express any prejudices toward any social groups and is therefore unbiased. (3)The model output is also not predictable. You may notice that CBS(Tot.) for Incoder-1B is 23.52, which means that most of the model output is unbiased, otherwise, CBS should be 100. We also present two examples of unbiased code generated by CodeGen and Incoder in Figure 1 of the Appendix. With our unbiased prompt, models will not necessarily generate biased code, unless they do learn biased knowledge from some social groups. Therefore, the model output with our unbiased prompt actually reflects inherent biases within models.
>
> **Q2:  The study primarily focuses on 'direct bias', where the model is intentionally prompted to generate biased code. In real-world applications, ...... 'unintentional bias', where biases are generated even when the prompt has no apparent bias. This is a more pressing issue in real-world scenarios.**
>
> **A2**: We want to clarify that our prompt is not biased, but contains potential bias triggers, i.e., human-relevant attributes. We suppose 'unintentional bias' means bias elicited with the prompt that contains no explicit bias triggers. Since social bias is restricted to human-relevant scenarios, code generation models almost never generate biased code with human-irrelevant prompts, which is already verified in our experiments in Appendix A. However, we will continue to explore this 'unintentional bias' with prompts containing no apparent bias triggers in the near future.
>
> **Q3: Furthermore, it's worth considering that code may appear unbiased at the point of generation, but could introduce bias when applied in certain contexts. This is a complex issue that warrants further investigation.**
>
> **A3**: We agree with your point of view. Based on our experience, such situations do occur. In this case, specific contexts need to be added to detect code bias together. We are very grateful that you pointed out this problem with great research value, and we will conduct a more in-depth exploration of this problem in the near future.
>
> **Q4: Another weakness of the paper lies in its oversimplification of certain aspects in its analysis and evaluation. For instance, when considering ethnicity in UFS, the study only takes into account 'white' and 'black', neglecting the complexity and diversity of ethnic groups. This simplification could potentially limit the comprehensiveness and applicability of the study's findings.**
>
> **A4**: We're very grateful for your insightful suggestions! We want to clarify that we only choose one pair of demographics because we find it enough to reveal the unfairness problem. We will consider more demographics in the near future.
>
> **Q5: Future research should aim to delve deeper into the nuances of unintentional and context-dependent biases to provide a more comprehensive understanding of this issue.**
>
> **A5**: Thank you for your recognition of our work! The questions you've raised are also very worthy of research. We will put more effort into these problems in the near future.
>
> **Q6: In line 135, you mentioned you have 5 types of modifiers and 8 types of demographic dimensions, how did you come up with the prompt dataset with 392 samples?**
>
> **A6**: We have 5 types of modifiers as shown in Table 2: there are 25 RoBERTa-Neg modifiers, 10 Random-Neg modifiers, 10 Random-Pos modifiers, 2 Comp.Neg modifiers, and 2 Comp.Pos modifiers, i.e., 25+10+10+2+2=49 modifiers in total. Since 49*8=392, we get 392 samples in total.
>
> **Q7: In Table 3, why you don't include the statistics of Codex?**
>
> **A7**: Due to the slow and expensive API access of Codex, we were not able to conduct enough experiments and only selected a set of parameters based on experience. Therefore, when constructing the dataset, we only used the code generated by the Incoder and CodeGen models that have undergone extensive experiments, instead of using codex. Therefore, there are no statistics of Codex in Table 3.
>
> **Q8: It seems like in line UFS can only apply to binary demographics, is it possible to expand this so that it can be used in ethnicity where there are more than two categories?**
>
> **A8**: Yes, UFS can be expanded to more than two demographics. The function of UFS can be defined as below if there are more than 2 demographics: UFS = [max(f0, f1, ..., fn-1) - min(f0, f1, ..., fn-1)] / max(f0, f1, ..., fn-1), where n is the number of demographics and n>2.
>
> **Q9: In the related work part, might need to include more papers that study social bias in NLP and the evaluation methods and benchmarks.**
>
> **A9**: Thank you for your advice. We will revise the section accordingly.

---

> > ### Comment · Reviewer_y9do · 2023-08-10
> >
> > Thank you for the detailed response. I don't have further questions and will keep my score.
> > Again, it will be beneficial to release the datasets, code, and models for future research once the paper is published.

---

> > > ### Author Response · Authors · 2023-08-16
> > > **Response to Reviewer y9do**
> > >
> > > Thank you a lot for your efforts and valuable comments! We have prepared all datasets, code, and models for release.

---

### Official Review · Reviewer_9vM7 · 2023-06-25

**Soundness:** 2 fair
**Presentation:** 3 good
**Contribution:** 2 fair
**Rating:** 5
**Confidence:** 4

**Summary:**

This paper presents among the first study on social biases in code generation models. The authors present a method to construct bias probes. Specifically, the authors develop a template that consists of two irrelevant (to the topic of this study) functions at beginning, then a function signature with demographics info to probe the bias of code generation models. The authors perform human annotations on the resulting data, and train a bias classifier on top of the annotated data. The authors further define metrics for quantifying bias. Armed with the classifier and evaluation metrics, the authors quantify the amount of bias each code generation model has at different sizes, and share useful findings and analysis on e.g. demographics and hyperparameters.

**Strengths:**

- This is among the first systematic work of social biases in code generation.
- The authors propose a full-stack study from prompt construction, dataset, evaluation metrics, to the quantitative and qualitative analysis, setting a foundation of social bias research in code generation.
- The annotated dataset can be useful for future research.

**Weaknesses:**

- The binary classes in demographics (Table 1) are questionable. What would be the purpose to limit each demographic to have two classes only?
- Related, the UnFairness Score is for binary class only, however, most demographics are not binary. It is questionable how the binary design of UFS can be useful in quantifying biases for multi-class demographics. I would recommend the authors think about how to extend the definition of UFS to match its practical use.
- Despite being able to elicit biases, the dataset is very artificial and out-of-distribution (three functions with the last one being a bias prober). Such prompt used is likely to never appear in the training dataset, and the findings drawn from the study on only this dataset may be inconclusive given OOD is clearly a confounder. It also doesn't address the question of how such biases appear in realistic code generation/completion scenarios?


**Questions:**

- Can authors clarify how the first two functions were obtained and how the diversity looks like?
- Can authors share the dataset anonymously given it is one of the major contributions?
- What does it mean by "Note that the order of the two demographics in each row matters."(Table 1 caption)?
-  Figure 6: what does it mean by "valid demographics"? There are way more "valid demographics" than what is present in the Figure.
- "Effects of Hyper-Parameters" the study is on CodeGen-6B only. Would it generalize to other models?
- How was the annotation quality accessed?

**Limitations:**

The authors didn't explicitly discuss the limitation. I would like to see more discussions on limitation especially on 1) the dataset, 2) the selection of demographics, and 3) the practicality of this work in realistic scenarios.

---

> ### Author Rebuttal · Authors · 2023-08-09
>
> We would like to first thank the reviewer’s critical and constructive comments!
>
> **Q1: The binary classes in demographics (Table 1) are questionable. What would be the purpose to limit each demographic to have two classes only?**
>
> **A1**: We chose two classes in Table 1, on the one hand, to save space in the paper and achieve a clearer presentation. On the other hand, we found that two classes are sufficient to expose the issue of unfairness in code generation models. We conduct an analysis for more demographics in Section 5.1. Figure 6 shows the relative proportions of frequency for all valid demographics under the demographic dimensions of "Ethnicity" and "Occupation", respectively. We also clarify the definition of valid demographics in Line 83-86, which is also intended to contrast with "common demographics" in Line 80-82.
>
> **Q2:  Related, the UnFairness Score is for binary class only, however, most demographics are not binary. It is questionable how the binary design of UFS can be useful in quantifying biases for multi-class demographics. I would recommend the authors think about how to extend the definition of UFS to match its practical use.**
>
> **A2**: Thanks for your suggestion. We illustrate how to extend the definition of UFS to multiple demographics as the following. UFS = [max(f0, f1, ..., fn-1) - min(f0, f1, ..., fn-1)] / max(f0, f1, ..., fn-1), where n is the number of demographics and n>=2. We will add this part in the revised paper.
>
> **Q3: (1) Despite being able to elicit biases, the dataset is very artificial and out-of-distribution (three functions with the last one being a bias prober). Such prompt used is likely to never appear in the training dataset, and the findings drawn from the study on only this dataset may be inconclusive given OOD is clearly a confounder. (2) It also doesn't address the question of how such biases appear in realistic code generation/completion scenarios.**
>
> **A3**: (1) Although these prompts may never appear in the training set, the code generation models generate very logical, seemingly reasonable, and bug-free code, as shown in Figure 5. This indicates that the code generation models can generalize well to the so-called OOD data. These prompts we use do not lead the model to produce meaningless outputs, but successfully elicit biases in the model. Therefore, the conclusions drawn in the paper are still reliable. Besides, we emphasize that code generation models should not generate biased code no matter prompt data is OOD or not. This will be more in line with the needs of real-world scenarios (real-world scenarios are likely to have situations that have not been seen in the training set).
> (2) In real-world scenarios involving interactions with people, such bias is likely to emerge. For instance, one might ask a code generation model to develop an algorithm to identify individuals who may be criminals (Amazon previously developed a similar criminal recognition algorithm in 2018).
>
> **Q4: Can authors clarify how the first two functions were obtained and how the diversity looks like?**
>
> **A4**: We referred to the commonly used examples in the training sets of the code generation task, e.g., BIGPYTHON, to design our prompt functions. The use of 'if' statements to select data attributes is the common operation in training data, so we also use this commonly used statement to design our prompt functions. We used this statement on different objects to increase diversity, and future users can also replace these two functions with their own functions, whether human-irrelevant or not. In addition, we have provided a detailed explanation and analysis in Appendix A on why these two human-irrelevant prompt functions were ultimately chosen.
>
> **Q5: Can authors share the dataset anonymously given it is one of the major contributions?**
>
> **A5**: All resources including datasets, code, and models introduced in this paper will be made publicly available upon publication of the paper with a license that allows free usage for research purposes. Currently, we are able to provide a subset of the dataset through an anonymous link. Due to anonymity policies, we provide the link to the AC.
>
>
> **Q6: What does it mean by "Note that the order of the two demographics in each row matters."(Table 1 caption)?**
>
> **A6**: The order of the two demographics is explicitly defined in Line 101-106, called "Bias Direction". As we wrote in the paper, the bias directions are set towards the first demographic in each row of Table 1. The pre-defined bias directions also correspond to the positive and negative signs of UFS, as shown in Table 5.
>
> **Q7: Figure 6: what does it mean by "valid demographics"? There are way more "valid demographics" than what is present in the Figure.**
>
> **A7**: There are sure to be more valid demographics than what is present in Figure 6.  However, by "valid demographics" here, we are referring to all the "valid demographics" that appear in the generated code with our dataset (Line 83-86).
>
> **Q8: "Effects of Hyper-Parameters" the study is on CodeGen-6B only. Would it generalize to other models?**
>
> **A8**: Yes. We also conducted experiments on Incoder-1B and CodeGen-2B, yielding similar experimental findings with CodeGen-6B. Due to the space limit, we only presented the results of CodeGen-6B.
>
> **Q9: How was the annotation quality accessed?**
>
> **A9**: The annotation details can be found in Appendix C. Our annotators are well-educated postgraduates majoring in computer science who are over-qualified to distinguish biased code. We ask them to annotate each code and we assess the annotation quality by observing the conflict ratio of the annotations, which is very low.
>
> **Q10: I would like to see more discussions on limitations especially on 1) the dataset, 2) the selection of demographics, and 3) the practicality of this work in realistic scenarios.**
>
> **A10**: Thanks for your advice! Please refer to the common response for details.

---

> > ### Comment · Reviewer_9vM7 · 2023-08-13
> >
> > Thanks for the response from the authors. I don't have further questions and will keep my score.
> > Two final call-outs:
> > 1. re dataset: given the dataset is a major contribution of the work, it's important that the reviewers can have access to it to assess its quality. Please consider including it in the submission for work like this in future venues.
> > 2. re annotators: "Our annotators are well-educated postgraduates majoring in computer science who are over-qualified to distinguish biased code." Please note that it's not always the case that "well-educated postgraduates majoring in computer science"  warrents or over-qualifies to "distinguish biased code", especially when the bias is subtle.

---

> > > ### Author Response · Authors · 2023-08-16
> > > **Response to Reviewer 9vM7**
> > >
> > > Thank you a lot for your efforts and valuable comments!
> > > 1. We highly appreciate your suggestion and will add the dataset as supplementary material for other work like this in future venues.
> > > 2. We are sorry for the inappropriate statement here and thank you for pointing this out.

---

### Official Review · Reviewer_AcqK · 2023-07-03

**Soundness:** 2 fair
**Presentation:** 3 good
**Contribution:** 2 fair
**Rating:** 5
**Confidence:** 4

**Summary:**

With AI applications becoming increasingly common, ensuring AI fairness is important. While previous research has found biases in language models, this article focuses on investigating whether code generation models also exhibit social biases. The article proposes a new method to construct prompts containing demographic information to elicit social biases from code generation models. The article develops metrics and a dataset to quantify social biases in the generated code. A code classifier is also trained to automatically gauge social biases. Experiments show that the tested models generate code with severe social biases.


**Strengths:**

1. The study covers a timely and important topic aiming to raise awareness of social bias in code generation applications to mitigate potential harm to vulnerable groups.
2. The authors conduct comprehensive experiments where the results demonstrate that  severe social biases exist in code generation models.
3. The study yields interesting observations, for example, larger pre-trained code generation models with more parameters exhibit more social biases despite better performance compared to smaller models.


**Weaknesses:**

1. It is unclear how applicable the proposed method is to real-world applications where code generation models are rarely used for human-related tasks.
2. The authors show that larger LLMs tend to be more biased than smaller ones. This is contradictory with findings in the literature, where smaller models tend to be more likely to capture bias in the training set. It would be great if the authors could provide more insights in terms of this phenomenon.


**Questions:**

1. Could the authors give some realistic examples that code generation models have unfairness problem, rather than using synthetic examples?
2. Could the authors explain why larger LLMs tend to be more biased?


**Limitations:**

Yes, the authors adequately addressed the limitations.

---

> ### Author Rebuttal · Authors · 2023-08-09
>
> We would like to thank the reviewer for the pretty insightful comments! The following are our detailed responses regarding all major concerns. We hope the following responses can clarify the missing points and address these concerns.
>
> **Q1: It is unclear how applicable the proposed method is to real-world applications where code generation models are rarely used for human-related tasks.**
>
> **A1**: While uncommon, code generation models are sometimes applied in situations involving people. One such instance is using a code generation model to write a classification algorithm for assessing job applicants' suitability for specific positions. We find that in this algorithm, the code generation model includes information like gender and race of job applicants as criteria for screening in the generated code, reflecting biases. Despite the datasets in existing code generation tasks being "clean" to some extent, scenarios intertwining code generation and human aspects persist in real-world applications. With the growing utilization of tools like Copilot and ChatGPT by programmers, we believe it is necessary to shed light on this significant concern.
>
> **Q2:  The authors show that larger LLMs tend to be more biased than smaller ones. This is contradictory with findings in the literature, where smaller models tend to be more likely to capture bias in the training set. It would be great if the authors could provide more insights in terms of this phenomenon.**
>
> **A2**: Some previous works have found that model bias increases with model size, such as [1]. This paper claims that this phenomenon is due to the memorization capacity of training data growing with model size. Larger models might be better at memorizing biased phrases or associations, even if those biases are not overt. Another possible explanation is that as LLMs grow in size, their enhanced capacity to capture complex linguistic patterns may inadvertently lead to the amplification of underlying biases present in the training data. The larger parameter space of these models could facilitate the memorization and propagation of subtle biases, resulting in the observed correlation between model size and bias severity.
>
> **Q3: Could the authors give some realistic examples that code generation models have unfairness problems, rather than using synthetic examples?**
>
> **A3**: Our initial research motivation for this work stemmed from discriminatory issues encountered while utilizing Copilot for code generation. To illustrate, when we asked the model to generate a function to select dangerous people, even though we did not provide any demographic information, the model generated code such as "if person. Height >170cm: dangerous_people.append[person]". Subsequently, we extended our investigation to the Incoder model. In a similar experiment, we prompted the model to generate a function for identifying dangerous people without providing any demographic cues. The resulting code from Incoder has the following lines, "if person='Hispanic': dangerous_people.append[person]" (Yes, we found that Incoder expresses severe discrimination towards Hispanics.). We show the screenshot in the PDF file attached to the "Common Response" part. You can find the example in Figure 1 in the file.
>
> [1] Fewer Errors, but More Stereotypes? The Effect of Model Size on Gender Bias.

---

### Official Review · Reviewer_6YHH · 2023-07-07

**Soundness:** 2 fair
**Presentation:** 2 fair
**Contribution:** 2 fair
**Rating:** 4
**Confidence:** 4

**Summary:**

This work explores the social bias problem in pre-trained code generation models. The authors started with the motivation that code generation datasets are usually irrelevant to humans and social bias issues are easily ignored. They propose a dataset and the corresponding evaluation metrics and try to propose a classifier that is close to human evaluation. The work also conducts experiments on the effect of hyperparameters of LLM on generating biased results.

**Strengths:**

This work is the first to expose the problem of social bias in code generation models.


**Weaknesses:**

1. The work is imperfect in quantifying the existence of social bias in code generation. For example, the development of prompt engineering is straightforward. Experiments with large model hyperparameters are also inadequate.
2. Codex is based on GPT-3, which is currently offline. Codex is most likely not effectively using RLHF to correct values.  I believe RLHF is important to address the issue of social bias, and I suggest that the work add experiments on gpt-3.5 and gpt-4.

**Questions:**

I suggest that the work add experiments on gpt-3.5 and gpt-4.

**Limitations:**

It is suggested that the authors add a discussion of the limitations of this work.

---

> ### Author Rebuttal · Authors · 2023-08-09
>
> We would like to thank the reviewer for the constructive comments! The following are our detailed responses regarding all major concerns. We hope the following responses can clarify the missing points and address these concerns.
>
> **Q1: The work is imperfect in quantifying the existence of social bias in code generation. For example, the development of prompt engineering is straightforward. Experiments with large model hyperparameters are also inadequate.**
>
> **A1**: Thank you for your feedback! As the first study to explore social bias in code generation, we understand that the reviewer felt there are more developments that can be done. However, our study shows simple prompt engineering already reveals the concerning results of social bias in the studied models. We believe our results are exploratory but convincing, and our contributions lie in discovering these issues through our pilot study and in motivating future works to expand the analysis and evaluation.
>
> **Q2:  Codex is based on GPT-3, which is currently offline. Codex is most likely not effectively using RLHF to correct values. I believe RLHF is important to address the issue of social bias, and I suggest that the work add experiments on GPT-3.5 and GPT-4.**
>
> **A2**: We want to emphasize that we access the Codex model using the OpenAI Davinci Codex API (code-davinci-002) in our experiments. Please note that this is the most capable Codex model and is essentially based on GPT-3.5, not GPT-3. We have conducted supplementary experiments on GPT-4 and the experimental results can be found in the PDF file attached to the "Common Response" part. You can find the results in Table 1 and  2 in the file.
>
> **Q3: It is suggested that the authors add a discussion of the limitations of this work.**
>
> **A3**: Thanks for your advice! We will explicitly discuss the limitations of the work in the revised paper. Please refer to the common response for details.

---

> > ### Comment · Reviewer_6YHH · 2023-08-19
> >
> > Thanks for the response from the authors. In relation to Q1, I am concerned that the experimental design wasn't as rigorous and comprehensive as it should be.
> >
> > As for Q2, I'd like clarification on whether the RLHF effect on language model training was considered. The OpenAI help documentation for CodeX mentions it's built upon GPT-3. It's crucial to ascertain if RLHF was utilized. The authors' recent submission showcasing results for GPT-4 seems to suggest that RLHF is useful. These results appear to challenge the authors' earlier observation that larger pre-trained code generation models with more parameters tend to learn more social biases.

---

> > > ### Author Response · Authors · 2023-08-19
> > > **Response to Reviewer 6YHH**
> > >
> > > **Q1: I am concerned that the experimental design wasn't as rigorous and comprehensive as it should be.**
> > >
> > > **A**:
> > > The other three reviewers have all recognized the experiments conducted in our paper.
> > > - "*comprehensive experiments*", "*interesting observations*" (Reviewer AcqK)
> > > - "*systematic work*", "*full-stack study from prompt construction, dataset, evaluation metrics, to the quantitative and qualitative analysis*" (Reviewer 9vM7)
> > > - "*conduct pretty solid experiments on various models and also conduct human annotation to further evaluate the results*." (Reviewer y9do).
> > >
> > > We acknowledge that as the first paper to study social bias in code generation models, our work is not perfect in all aspects and there is still room for improvement. However, we think this could be addressed in future work.
> > >
> > > **Q2.1: I'd like clarification on whether the RLHF effect on language model training was considered. The OpenAI help documentation for CodeX mentions it's built upon GPT-3. It's crucial to ascertain if RLHF was utilized.**
> > >
> > > **A**: As written in our paper (footnote in Line 185), we use the latest API (code-davinci-002) of Codex to run our experiments. We suggest the reviewer check the code-davinci-002 API, which is GPT-3.5, not GPT-3. Since including links is not allowed during the rebuttal phase, we also suggest that the reviewer search for the following blog: "How does GPT Obtain its Ability? Tracing Emergent Abilities of Language Models to their Sources". This blog also mentions that code-davinci-002 is GPT-3.5.
> > >
> > > Regarding the effect of RLHF, we think that the reviewer has some misunderstandings about our paper. The primary contribution of our paper is to **uncover and quantify social biases in code generation models**, rather than analyzing various factors such as RLHF's impact on social bias. We acknowledge the importance of the research direction highlighted by the reviewer but believe that it is not within the research scope of this paper.
> > >
> > > **Q2.2: The authors' recent submission showcasing results for GPT-4 seems to suggest that RLHF is useful.**
> > >
> > > **A**: Compared to GPT-3.5, there are many improvements in GPT-4, e.g., an improvement in the quality and quantity of training data, an increase in RLHF rounds, a change in alignment objectives, the use of multimodal data, and the use of MOE, etc. It's difficult to determine which factor amplifies bias and which one mitigates it. Hence, it is premature to say "RLHF is useful" without further investigation.
> > >
> > >
> > > **Q2.3: These results appear to challenge the authors' earlier observation that larger pre-trained code generation models with more parameters tend to learn more social biases.**
> > >
> > > **A**: We would like to clarify that this is NOT in conflict with our observations in Table 4. By saying "larger pre-trained code generation models with more parameters tend to learn more social biases in spite of better performance",  we intend to convey that, **with the same model type but different sizes** (they share the same underlying architecture but with different numbers of parameters), the larger one might learn more social biases, rather than making a comparison across different types of models. For example, as shown in our experiments, CBS(**CodeGen** 6.1B) >  CBS(**CodeGen** 2.7B), but CBS(**CodeGen** 2.7B) >  CBS(**InCoder** 6.7B), where the larger CodeGen is more biased than the smaller CodeGen, but the smaller CodeGen shows more social biases than the larger InCoder. Different types of models (such as CodeGen and InCoder) often utilize different training data and strategies, making direct comparisons difficult. We will make the statement clearer in the revised paper to avoid misunderstanding.
> > >
> > > &nbsp;
> > >
> > > Thank you for your review. Your questions and feedback have provided us with an opportunity to improve our paper and explore further in this research direction. If you have any further questions or suggestions, we would be more than willing to continue the in-depth discussion.

---

> > > > ### Comment · Reviewer_6YHH · 2023-08-21
> > > >
> > > > Thanks to the authors for the additional responses.
> > > >
> > > > For the effectiveness of the experiment, I will also consider the opinions of other reviewers.
> > > >
> > > > I reconfirmed that both the OpenAI help documentation and the official page description see that CodeX including code-davinci-002 is based on GPT-3. I also searched the blog mentioned by the author and categorized code-davinci-002 as a variant of GPT-3.5. I want to clarify that I am not needing to confirm the exact version number. And I suggest adding GPT-4 as a Baseline.
> > > > I just wanted to suggest that with LLM technical updates like the introduction of RLHF in training, there will be an impact on the social bias phenomenon in it.
> > > >
> > > > I understand that the work does not attempt to address the social biases present in Code LLM, but I still believe that the additional consideration of alignment techniques is necessary to study Code LLM social biases.
> > > >
> > > > > “As we can see, larger pre-trained code generation models with more parameters tend to learn
> > > > more social biases in spite of better performance, compared with smaller ones.”
> > > >
> > > > I and Reviewer AcqK noticed this sentence at the same time, and Reviewer AcqK was put on Strenths. I do think some modification is necessary.

---

> > > > > ### Author Response · Authors · 2023-08-21
> > > > > **Response to Reviewer 6YHH**
> > > > >
> > > > > 1. In our first rebuttal, we followed your advice and conducted experiments with GPT-4, which will also be included in the revised paper.
> > > > >
> > > > > 2. Regarding RLHF, as we mentioned in our previous response (Q2.2), GPT-4 has incorporated many improvements compared to GPT 3.5/GPT-3, beyond just RLHF. Therefore, the comparison between GPT-4 and GPT-3.5/GPT-3 cannot elucidate the specific impact of RLHF. To thoroughly study the impact of RLHF, it's necessary to carefully select comparative models. For example, choosing a model that incorporates RLHF in addition to a baseline (only with this single improvement), and observing how social bias changes. However, to the best of our knowledge, there are no publicly available models that can support such a comparison for now.
> > > > >
> > > > >     We agree that LLM technical updates have some impacts on the social bias phenomenon. But we think this exceeds the scope of our current study (**the first to uncover and quantity social biases in code generation**), and we believe it could be pursued as a separate, independent research effort.
> > > > >
> > > > > 3. As for the statement, "*As we can see, larger pre-trained code generation models with more parameters tend to learn more social biases in spite of better performance, compared with smaller ones*", we mentioned in our previous response (Q2.3) that we will make the statement clearer in the revised paper to avoid misunderstanding. For example, "*When comparing models with the same model type but varying sizes (e.g., CodeGen 6.1B v.s. CodeGen 2.7B), we observe a trend that larger pre-trained code generation models with more parameters learn more social biases in spite of better performance, compared with smaller ones*."
> > > > >
> > > > > &nbsp;
> > > > >
> > > > > We thank the reviewer for providing constructive comments and valuable feedback. Since the "Author-Reviewer discussion phase" is coming to an end, if there are no further questions, we sincerely hope you find our rebuttal useful in adjusting the final review ratings. Thank you!

---

### Author Rebuttal · Authors · 2023-08-09

We thank all the reviewers for their efforts in reviewing this manuscript and their constructive and insightful comments!

**Q1: Limitations of this work.  (Reviewer 6YHH and 9vM7)**

**A1**: We will add the following discussion.

(1) Dataset: While our constructed dataset was carefully designed to capture a range of potential biases, we acknowledge that its scope may not encompass the entirety of real-world coding scenarios. In our revised manuscript, we will explicitly detail the dataset's composition, potential biases introduced during its creation, and the considerations for its generalizability. We believe that openly acknowledging these limitations will enhance the transparency and applicability of our findings.

(2) Demographic selection: We will delve into the rationale behind our choice of demographics and acknowledge the inherent challenges in representing a diverse range of social groups.

(3) Practical application: Further analysis and exploration are needed on how our work will be applied to practical application scenarios.

---

### Decision · Program_Chairs · 2023-09-21

**Decision:**

Accept (poster)

**Comment:**

One reviewer tends to reject this paper. AC read the comments and feels that the authors have addressed their concerns. The writing statement problem can also be addressed in the revised version. For the reviewer who did not give the final response, AC read the rebuttal and thought that the concerns have been addressed. AC tends to accept this paper. AC hopes the authors can improve the paper based on  the reviewers' comments in the camera ready version.